# Clinical and Molecular Characterization of Classical-Like Ehlers-Danlos Syndrome Due to a Novel *TNXB* Variant

**DOI:** 10.3390/genes10110843

**Published:** 2019-10-25

**Authors:** Daisy Rymen, Marco Ritelli, Nicoletta Zoppi, Valeria Cinquina, Cecilia Giunta, Marianne Rohrbach, Marina Colombi

**Affiliations:** 1Connective Tissue Unit, Division of Metabolism and Children’s Research Centre, University Children’s Hospital, 8032 Zürich, Switzerland; Cecilia.Giunta@kispi.uzh.ch (C.G.); Marianne.Rohrbach@kispi.uzh.ch (M.R.); 2Division of Biology and Genetics, Department of Molecular and Translational Medicine, University of Brescia, 25123 Brescia, Italy; marco.ritelli@unibs.it (M.R.); nicoletta.zoppi@unibs.it (N.Z.); valeria.cinquina1@unibs.it (V.C.); marina.colombi@unibs.it (M.C.)

**Keywords:** Tenascin X, TNXB, Ehlers-Danlos syndrome, EDS, connective tissue, collagen

## Abstract

The Ehlers-Danlos syndromes (EDS) constitute a clinically and genetically heterogeneous group of connective tissue disorders. Tenascin X (TNX) deficiency is a rare type of EDS, defined as classical-like EDS (clEDS), since it phenotypically resembles the classical form of EDS, though lacking atrophic scarring. Although most patients display a well-defined phenotype, the diagnosis of TNX-deficiency is often delayed or overlooked. Here, we described an additional patient with clEDS due to a homozygous *null*-mutation in the *TNXB* gene. A review of the literature was performed, summarizing the most important and distinctive clinical signs of this disorder. Characterization of the cellular phenotype demonstrated a distinct organization of the extracellular matrix (ECM), whereby clEDS distinguishes itself from most other EDS subtypes by normal deposition of fibronectin in the ECM and a normal organization of the α5β1 integrin.

## 1. Introduction

The Ehlers-Danlos syndromes (EDS) constitute a clinically and genetically heterogeneous group of connective tissue disorders. Patients present with joint hypermobility, skin hyperextensibility and tissue fragility, giving rise to easy bruising and atrophic scarring. Although most EDS are caused by mutations in genes coding for the fibrillar collagens or collagen-modifying enzymes, over the last two decades, several disorders due to defects in other components of the extracellular matrix (ECM) have been delineated [1]. 

Tenascins comprise a family of glycoproteins, which modulate the adhesion of cells to their ECM. Tenascin X (TNX) is ubiquitously expressed, but the highest levels are found in muscle and loose connective tissue [2]. TNX is thought to regulate fibril spacing by direct binding to the distinct collagen fibrils in the ECM or by indirect binding via decorin [3]. Additional roles for TNX in elastic fiber remodeling and regulating the expression of certain ECM components, e.g., collagen VI, proteoglycans and matrix metalloproteases have been suggested [4,5,6,7].

The clinical relevance of TNX-deficiency was first proposed in 1997 by the identification of a patient presenting both congenital adrenal hyperplasia and an EDS-like phenotype. Genetic analysis revealed a contiguous deletion of the partially overlapping genes *CYP21B* and *TNXB* on chromosome 6. Although a TNX *null*-phenotype was verified both on the protein and on the mRNA level, a mutation on the second allele could not be identified [8]. In 2001, Schalkwijk et al. demonstrated that isolated TNX-deficiency resulted in an autosomal recessive form of EDS, resembling the classical type, however lacking atrophic scarring [9]. In 2017, TNX-deficiency was officially classified as “classical-like EDS” (clEDS), with generalized joint hypermobility, hyperextensible, soft and/or velvety skin without atrophic scarring and easy bruising being the typical clinical hallmarks of the disorder [1].

In 2002, Mao et al. developed a TNX-deficient mouse model, mimicking the EDS phenotype. Indeed, as observed in humans, *Tnxb^-/-^* mice were morphologically normal at birth, but displayed progressive hyperextensibility of the skin over time. The group showed that the phenotype did not relate to aberrant collagen fibrillogenesis, but was rather due to altered deposition, and therefore, reduced density of collagen in the ECM [10].

Although most TNX-deficient patients display a well-defined clinical phenotype, the diagnosis is often delayed or overlooked. The former is attributed to the molecular analysis of the *TNXB* gene being complicated by the presence of a highly homologous pseudogene and to the fact that the measurement of TNX in serum is not widely available [11]. The latter is mainly caused by poor clinical awareness, which unfortunately applies to many rare disorders.

Here, we reported on an additional patient with clEDS and a novel homozygous disease-causing variant in *TNXB* to further elaborate the clinical phenotype. Furthermore, we reviewed the clinical features of the clEDS patients described to date, in order to create a well-defined description of the phenotype and increase clinical awareness.

## 2. Materials and Methods

### 2.1. Ethical Compliance

This study is in accordance with the Helsinki declaration and its following modifications. Ethics approval has been granted (KEK Ref.-Nr. 2014-0300 and Nr. 2019-00811) in the presence of a signed informed consent of the patient for genetic testing, skin biopsy and the publication of clinical pictures. The patient was evaluated at the University Children’s Hospital of Zürich. Targeted next-generation sequencing (NGS) panel for 101 connective tissue disorders (Appendix A) was performed at the Institute of Medical Genetics of the University of Zürich. *TNXB* mutational screening by Sanger sequencing and Multiplex Ligation-dependent Probe Amplification (MLPA) was achieved at the Division of Biology and Genetics, Department of Molecular and Translational Medicine, of the University of Brescia. 

### 2.2. Cell Culture

As part of the diagnostic workup of EDS, a punch biopsy of the patient’s skin for establishing fibroblast cultures for collagen biochemical analysis was previously obtained. The biological material was stored in the Biobank of the Division of Metabolism at the Children’s Hospital Zürich. Fibroblasts from the patient and from sex- and age-matched healthy individuals were routinely maintained at 37 °C in a 5% CO_2_ atmosphere in Earle’s Modified Eagle Medium (MEM) supplemented with 2 mM L-glutamine, 10% FBS, 100 μg/ml penicillin and streptomycin (Life Technologies, Carlsbad, CA, USA). Fibroblasts were expanded until full confluency and then harvested by 0.25% trypsin/0.02% EDTA treatment at the same passage number.

### 2.3. Molecular Analysis

Mutational screening was performed on genomic DNA purified from peripheral blood leukocytes using standard procedures. In particular, all exons and their intron-flanking regions of the *TNXB* gene (NM_0019105.7, NP_061978.6) were PCR amplified with the GoTaq Ready Mix 2X (Promega, Madison, WI, USA) by using optimized genomic primers that were analyzed for the absence of known variants using the GnomAD database (https://gnomad.broadinstitute.org/). For the pseudogene-homolog region (exons 32–44), Sanger sequencing was performed by nested PCR, using a *TNXB*-specific long-range PCR product encompassing the 3’-end of *TNXB* as a template (for details on primer sequences and PCR conditions see Appendix A). PCR products were purified with ExoSAP-IT (USB Corporation, Cleveland, OH, USA), followed by bidirectional sequencing with the BigDye Terminator v1.1 Cycle Sequencing kit on an ABI3130XL Genetic Analyzer (Thermo Fisher Scientific, South San Francisco, CA, USA). The sequences were analyzed with the Sequencher 5.0 software (Gene Codes Corporation, Ann Arbor, MI, USA) and variants were annotated according to the Human Genome Variation Society (HGVS) nomenclature with the Alamut Visual software version 2.11 (Interactive Biosoftware, Rouen, France). Deletion/duplication analysis of *TNXB* was performed using the MLPA assay P155, according to manufactures’ instructions (MRC-Holland, Amsterdam, the Netherlands).

### 2.4. RNA Extraction and Quantitative Real-Time PCR

Total RNA was purified from skin fibroblasts of the patient and 3 healthy individuals using the Qiagen RNeasy kit, according to the manufacturer’s instructions (Qiagen, Hilden, Germany). RNA quality control was performed on an Agilent 2100 Bioanalyzer (Agilent Technologies, Santa Clara, CA, USA). Relative expression levels of the *TNXB* transcript were analyzed by quantitative real-time PCR (qPCR). Of the total RNA, 3 μg was reverse-transcribed with random primers by a standard procedure. qPCR reactions were performed in triplicate with the SYBR Green qPCR Master Mix (Thermo Fisher Scientific, South San Francisco, CA, USA), 10 ng of cDNA, and with 10 μM of each primer set by using the ABI PRISM 7500 Real-Time PCR System with standard thermal cycling conditions. The *HPRT*, *GAPDH* and *ATP5B* reference genes were also amplified for normalization of cDNA loading. Relative mRNA expression levels were normalized to the geometric mean of these reference genes and analyzed using the 2^−(ΔΔCt)^ equation. Amplification plots, dissociation curves and threshold cycle values were generated by ABI Sequence detection system software version 1.3.1. Statistical analyses were performed with the GraphPad Prism software (San Diego, CA, USA). The results were expressed as the mean values of relative quantification ± Standard Error of the Mean (SEM). Statistical significance between groups was determined using one-way ANOVA. *P*-values were corrected for multiple testing using the Tukey’s method.

### 2.5. Collagen Biochemical, Ultrastructural and Immunofluorescence Microscopy Studies

Collagen steady state analysis in patient’s cultured fibroblasts was conducted prior to genetic analysis to assess possible anomalies in collagen biosynthesis and secretion, as previously described [12]. Briefly, radioactively labeled and pepsinized procollagens from the patient and one healthy individual were separated on a 5% Sodium Dodecyl Sulphate Polyacrylamide Gel Electrophoresis (SDS-PAGE) and visualized by autoradiography. 

To analyze ECM organization of collagen type I (COLLI), collagen type III (COLLIII), collagen type V (COLLV) and fibronectin (FN), skin fibroblasts derived either from the patient or an unrelated control individual were grown for 72 hours in the presence of 50 μM ascorbic acid (for COLLI only), refed after 24 hours and immunoreacted as previously reported [13,14,15]. In brief, cold methanol fixed fibroblasts were immunoreacted with 1:100 anti-COLLI, anti-COLLIII, anti-COLLV (Millipore-Chemicon Int., USA), and anti-FN (Sigma Chemicals, St. Louis, MO, USA) antibodies (Ab). The ECM of TNX was investigated on methanol-fixed cells immunoreacting with 2 μg/ml anti-TNX Ab (Santa Cruz Biotec. Inc., USA). For the analyses of the α2β1, α5β1 and αvβ3 integrin receptors, cells were fixed in 3% paraformaldehyde (PFA)/60 mM sucrose and permeabilized with 0.5% Triton X-100 as previously reported [13,14,16]. In particular, fibroblasts were reacted for 1 hour at room temperature with 4 μg/ml anti-α5β1 (clone JBS5), anti-α2β1 (clone BHA.2) and anti-αvβ3 (clone LM609) integrin monoclonal Abs (Millipore-Chemicon Int., USA). The cells were then incubated for 1 h with Alexa Fluor 488 anti-rabbit and Alexa Fluor 555 anti-mouse Ab (Thermo Fisher Scientific, South San Francisco, CA, USA), or with rhodamine-conjugated anti-goat IgG (Calbiochem-Merck, Germany). Immunofluorescence (IF) signals were acquired by a black-and-white CCD TV camera (SensiCam-PCO Computer Optics GmbH, Germany) mounted on a Zeiss fluorescence Axiovert microscope and digitalized by Image Pro Plus software (Media Cybernetics, Silver Spring, MD, USA). 

## 3. Results

### 3.1. Case Report

The patient was referred to our center at the age of 41 years because of a suspected connective tissue disorder. She was the first child of consanguineous parents of Swiss origin. The family history was unremarkable. The patient displayed a congenital dislocation of the left hip. Due to recurrent dislocations, an arthrodesis was performed at the age of 2 years. Immediately after surgery, a fracture of the left femur occurred, requiring osteosynthesis. Over the years, multiple joint dislocations ensued, mainly of the shoulders. During childhood, the patient developed a progressive kyphoscoliosis, which was left untreated. At present, the patient displays a right sided gibbus deformity (Figure 1). Upon radiography of the spine, a Cobb angle of 22, 38 and 15 degrees was measured at the cervical, thoracic and lumbar level, respectively. 

In addition, the patient presented a progressive arthrosis of the knee and degenerative changes of the spine. Bone densitometry at the age of 41 years was normal. Generalized joint hypermobility was absent (Beighton score 3/9). The patient reported mild muscular weakness. Computed tomography of the abdomen demonstrated fatty degeneration of the abdominal, gluteal and pelvic floor musculature. The patient was noted to have a soft, dough-like skin texture associated with skin hyperextensibility. In addition, the skin was thin and translucent with multiple varicose veins and edema of the lower extremities (Figure 2c). Redundant and sagging skin was found on both knees (Figure 2a). Striae were already present at young age. Feet and hands were short and broad, with the presence of brachydactyly (Figure 2d,g). Acrogeria of the hands was noticed, with an increased amount of palmar skin creases (Figure 2e). Several subcutaneous nodules were present at the interphalangeal joints of the fingers. Piezogenic papules were present on both feet (Figure 2f). While standing upright, the subcutaneous tissue of the feet was pushed aside, leading to neurogenic pain and the inability to walk barefoot or to go longer distances (Figure 2f). The patient reported large hematomas to occur after minimal trauma. Atrophic scarring was not present (Figure 2b). However, wound healing was clearly delayed with wound dehiscence after surgery. The patient presented with an inguinal and umbilical hernia at the age of 27 and 36 years, respectively, requiring surgical correction. The patient mothered three children, all born through C-section. A urogenital or rectal prolapse did not occur. The patient did, however, suffer from hemorrhoids. Colonoscopy at the age of 40 years demonstrated asymptomatic diverticulosis of the sigmoid colon. The patient presented with one episode of ileitis of unknown origin. In addition, the patient had two episodes of dactylitis and suffers from recurrent aphthous stomatitis. No biochemical indication of an underlying inflammatory disease was found. At the age of 42 years, a spontaneous perforation of the small bowel occurred, necessitating surgery and resection of the involved bowel loop. The intestinal specimen obtained during surgery was reported to be very fragile. An echocardiography at the age of 41 years demonstrated an intact mitral valve and normal dimensions of the aorta. The patient was known with a mild myopia. Hearing was normal to date.

### 3.2. Molecular and Biochemical Findings

#### 3.2.1. Molecular Analysis

Previous molecular investigations of the patient, all with negative results, included molecular genetic analysis of *COL1A1* and *COL1A2* on cDNA for the suspicion of arthrochalasis EDS, and an NGS panel for 101 connective tissue disorders. Given that the *TNXB* gene was not present in the NGS panel and considering the clinical presentation of the patient that was suggestive for clEDS, we sequenced *TNXB* by traditional Sanger method. Sequencing of all exons and exon-intron boundaries of *TNXB* (NM_0019105.7, NP_061978.6) revealed the homozygous c.5362del, p.(Thr1788Profs*100) variant in exon 15 (Figure 3a), which was absent in all public variant databases. Hence, the novel pathogenic *TNXB* variant (identifier #0000591583) was submitted to the Leiden Open Variation Database (LOVD, https://databases.lovd.nl/shared/genes/TNXB).qPCR analysis on cDNA obtained from patient’s dermal fibroblast showed that the c.5362del transcript, which leads to frameshift and formation of a premature termination codon (PTC) p.(Thr1788Profs*100), represents a null-allele undergoing nonsense-mediated mRNA decay (Figure 3b). Consistently, IF analysis on the patient’s fibroblasts with a specific Ab against TNX demonstrated a complete absence of the protein in the ECM (Figure 3c).

#### 3.2.2. Collagen Biochemical and Immunofluorescent Analysis

To assess the effects on collagen biosynthesis and secretion, we performed collagen steady-state and pulse-chase analyses in dermal fibroblasts from the patient. Normal relative proportions of COLLI, COLLIII and COLLV were present in the cell lysate and in the medium (Appendix A). There were no indications for altered modification of the different collagens.

To investigate the effect of TNX-deficiency on COLLI-, COLLIII-, COLLV- and FN-ECM organization and on the expression of their integrin receptors, dermal fibroblasts from the patient and one healthy individual were investigated by IF (Figure 4). Control fibroblasts organized a reticular ECM of COLLIII and COLLV, rare fibrils of COLLI, an abundant FN-ECM and expression of the canonical α2β1 and α5β1 integrins, whereas patient’s fibroblasts showed a lack of organized COLLI, COLLIII, and COLLV in the ECM, though proteins were present at different levels inside the cells. The disorganization of the COLLs-ECM is associated with a strong reduction of their canonical α2β1 integrin receptor (Figure 4a). Contrarily, the FN-ECM organization and the expression of its canonical integrin receptor α5β1 were not affected in TNX-deficient cells. Therefore, the alternative FN-receptor αvβ3 was almost undetectable both in control and patient cells (Figure 4b).

## 4. Discussion

Most known EDS subtypes are autosomal dominant inherited disorders due to defects in genes coding for the fibrillar collagens or collagen-modifying enzymes. TNX-deficiency differs from the more frequent subtypes not only in its autosomal recessive inheritance pattern but also in the fact that it is caused by a defect in an ECM component other than collagen. The presumed role of TNX in the stabilization and maturation of the collagen- and elastin-ECM is reflected in the cellular and clinical phenotype of the patients described to date [2,3,4,5,6,7]. Similar to the data in *Tnxb^-/-^* mice, we could demonstrate that collagen biosynthesis and secretion were unaffected in our patient’s fibroblasts, whereas a disarray of the COLLI-, COLLIII- and COLLV-ECM and a strong reduction of their canonical α2β1 integrin receptor was observed in vitro (Figure 4a) [10]. Unlike the typical EDS cellular phenotype, our patient’s fibroblasts were characterized by a normal organization of the FN-ECM and therefore a normal expression of its canonical integrin receptor α5β1 (Figure 4b) [10,17].

Thus far, about 30 patients with complete TNX-deficiency have been described in literature [8,9,18,19,20,21,22,23,24,25,26]. Most patients displayed either a complete lack of TNX in serum or biallelic mutations in *TNXB* leading to nonsense-mediated mRNA decay (Appendix A). As expected, those presenting missense mutations had milder or late-onset clinical manifestations [17,24].

As its name implies, TNX-deficiency or classical-like EDS (clEDS) phenotypically resembles the classical form of EDS (cEDS), with the triad of soft/velvety hyperextensible skin (20/20), generalized joint hypermobility (15/19) and a varying degree of tissue fragility (20/20) as its main clinical features (Table 1) [9]. Unlike cEDS, atrophic scarring is absent in patients with clEDS. However, about 50% of TNX-deficient patients present delayed wound healing with wound dehiscence upon suture removal and widened scars (Figure 2b). Data in *Tnxb^-/-^* mice suggest that, contrary to cEDS, abnormal wound healing in clEDS is not due to altered matrix deposition in the early phases of wound healing, but is rather caused by decreased ECM stabilization and maturation during the later stages. Indeed, contrary to the high levels of *TNXB* in normal skin, its expression during the early phase of wound healing is non-existing and only increases over time [27].

Whereas joint hypermobility seems to decrease with age, recurrent dislocations of the large joints (18/19) remain a problem over time and can be considered as the most frequent debilitating finding in the TNX-deficient patients described to date (Table 1). 

Interestingly, several patients display deformities of the hands and feet, most of which can be ascribed to the underlying connective tissue weakness, such as pes planus, hallux valgus, piezogenic papules and acrogeric hands. Conversely, the presence of brachydactyly and broad hands and feet point to a role of TNX in development [28]. In addition, our patient presented chronic venous insufficiency, varicose veins and non-pitting ankle edema, which has thus far been observed in almost 30% of the TNX-deficient population (Table 1). The venous insufficiency and its consequences might be explained by the aberrant elastic fiber remodeling observed in TNX-deficient dermal fibroblasts and possible changes in endothelial cell proliferation [5,29,30]. Indeed, the interaction between TNX and vascular endothelial growth factor B (VEGF-B) is known to stimulate endothelial cell proliferation [30,31].

Bone is considered to be a target organ of osteogenesis imperfecta rather than EDS. However, premature osteopenia or osteoporosis have been published in various types of EDS and have been mainly attributed to abnormal COLLI fibrillogenesis and ECM deposition [32]. Although bone loss due to an increased number of multinucleated osteoclasts has been found in *Tnxb^-/-^* mice, thus far, indications for skeletal fragility in clEDS have not been reported [33]. Moreover, in our patient, bone densitometry at the age of 41 years was normal.

Over time, neuromuscular symptoms have been increasingly recognized in various types of EDS, ranging from muscle weakness to myalgia and easy fatigability [21]. Indeed, a normal composition of the ECM and intact innervation are important for adequate functioning of the muscle [34]. Considering the cellular phenotype of clEDS, neuromuscular involvement is not surprising. Collagen VI, which is deficient in two myopathies, i.e., Ullrich congenital muscular dystrophy and Bethlem myopathy, is downregulated, although with conserved organization of the ECM, in as well TNX-*null* fibroblasts and *Tnxb^-/-^* mice [7,34,35]. In addition, TNX is widely distributed in peripheral nerve and its deficiency causes axonal polyneuropathy in almost 40% of the patients (Table 1) [21]. Our patient had a low muscular mass on clinical examination and reported distal muscle weakness. Computed tomography of the abdomen demonstrated fatty degeneration of the muscles as an incidental finding, a well-known phenomenon in myopathies of various origins. Interestingly, the neuromuscular phenotype seems to be mild or absent in children and progresses with age. 

Extra-articular symptoms in clEDS, such as gastrointestinal and cardiovascular complications, do not seem to be associated with certain mutations, but are rather related to age and have been reported after the 3rd–4th decade [18,22,23,25]. Gastrointestinal manifestations, although rare, can lead to life-threatening situations. Spontaneous perforation of hollow organs is a complication mainly seen in patients with vascular EDS (vEDS) due to mutations in COLLIII and has only sporadically been described in other EDS subtypes. Compared to other non-vEDS, a higher prevalence seems to exist in clEDS, ranging from tracheal or esophageal rupture to spontaneous perforation of colon, diverticula or small intestine [18,20,22,23,25,36]. Therefore, patients and treating physicians should be aware of this risk in order to minimize the occurrence of iatrogenic perforation and to not delay the diagnosis and treatment of spontaneous events. Contrary to patients with vEDS, aortic root dilatation or aneurysms are not common in clEDS, and have only been described in one non-published case [23]. Peeters et al. suggested that the co-expression of tenascin-C (TNC) in large blood vessels might compensate for the TNX-deficiency, giving rise to a normal arterial vessel wall [37,38]. Conversely, mitral valve involvement (i.e., mitral valve prolapse, thickening and/or insufficiency) has thus far been reported in 20% of clEDS patients, highlighting the need for cardiovascular follow-up (Table 1).

## 5. Conclusions

Although TNX-deficiency phenotypically resembles cEDS, absent atrophic scarring, the presence of short broad feet, brachydactyly, edema of the lower extremities, acrogeria or the occurrence of hollow organ perforation should initiate targeted diagnostics for clEDS, either by measuring the TNX concentration in serum or by mutation analysis of the *TNXB* gene.

## Figures and Tables

**Figure 1 genes-10-00843-f001:**
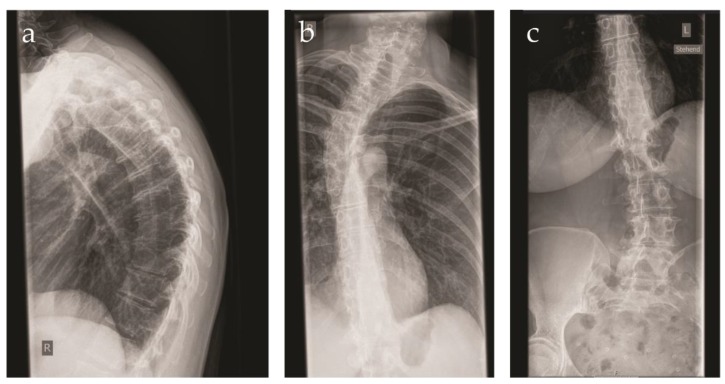
X-ray images of the vertebral column of the patient at the age of 41 years. (**a**) Side view demonstrating the pronounced kyphosis; (**b**) close-up of the pronounced cervical and thoracic scoliosis; (**c**) close-up of the lumbar scoliosis.

**Figure 2 genes-10-00843-f002:**
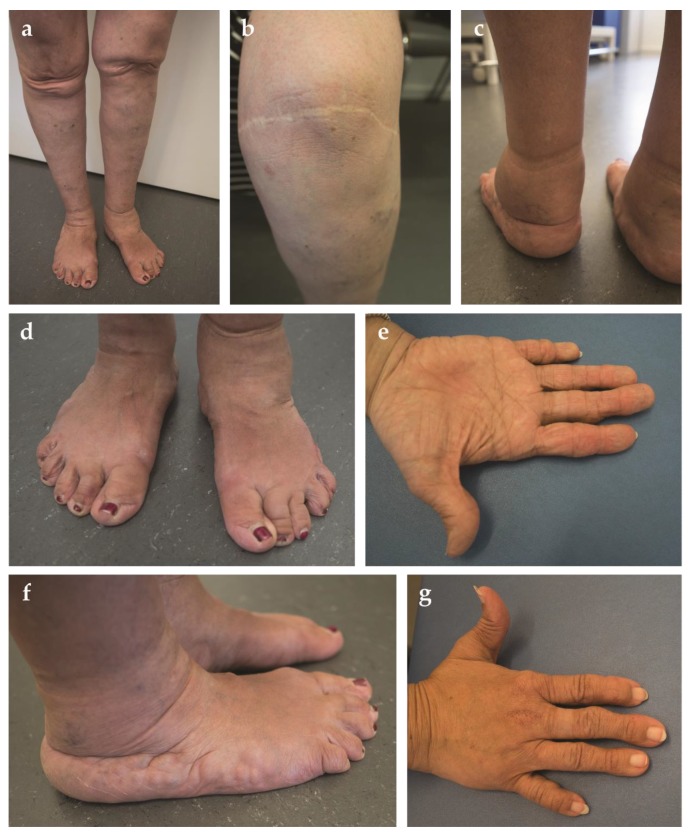
Clinical features of the patient. (**a**) Redundant and sagging skin on both knees; (**b**) widened scar on the left knee after surgery. No atrophic features are present; (**c**) chronic ankle edema in the absence of cardiac failure; (**d**) plane flat feet with broad forefeet and short digits; (**e**) increased palmar skin creases; (**f**) piezogenic papules; (**g**) short and broad acrogeric hands. Note the hyperextensibility of the thumb.

**Figure 3 genes-10-00843-f003:**
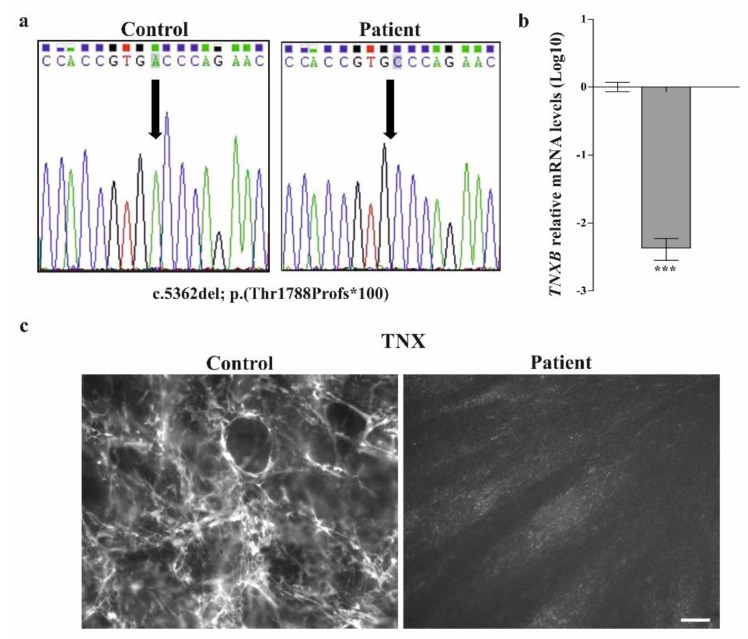
The *TNXB* c.5362del pathogenic variant causes nonsense-mediated mRNA decay and absence of the tenascin X protein in the ECM. (**a**) Sequence chromatograms showing the position of the novel c.5362del; p.(Thr1788Profs*100) variant (arrow) identified in homozygosity in exon 15 of the *TNXB* gene by Sanger sequencing; (**b**) qPCR analysis on cDNA obtained from patient’s dermal fibroblasts showed that the transcript with the p.(Thr1788Profs*100) frameshift variant undergoes nonsense-mediated mRNA decay. The relative quantification (RQ) of the *TNXB* transcript levels was determined with the 2^−(ΔΔCt)^ method normalized with the geometric mean of the *HPRT*, *GAPDH* and *ATP5B* reference genes. Bars represent the mean ratio of target gene expression in patients’ fibroblasts compared to three unrelated healthy individuals. qPCR was performed in triplicate, and the results are expressed as means ± SEM. The relative mRNA level of *TNXB* in the patient versus controls (about 142-fold decrease) is expressed as Log10 transformed value. Statistical significance (*** P < 0.001) was calculated with one-way ANOVA and the Tukey post hoc test. (**c**) IF analysis on patient’s skin fibroblasts with a specific antibody against TNX showing the absence of the protein in the ECM compared to control cells. Scale bar: 10 μm.

**Figure 4 genes-10-00843-f004:**
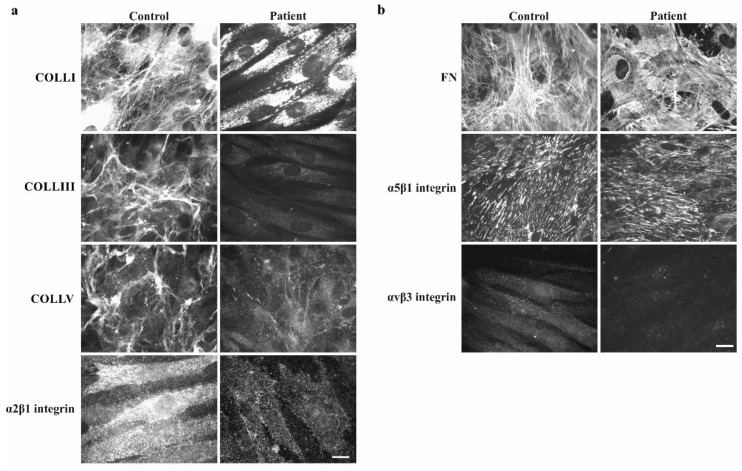
TNX-deficiency leads to the disarray of collagens type I, type III and type V in the ECM together with the reduction of the α2β1 integrin. (**a**) IF analysis with specific Ab demonstrates that TNX-deficient cells are characterized by the lack of organized COLLI, COLLIII and COLLV in the ECM, though proteins are present at different levels inside the cells. The disorganization of the COLLs-ECM is associated with a strong reduction of their canonical α2β1 integrin receptor. (**b**) TNX-deficient cells deposit FN in the ECM and express its canonical integrin receptor α5ß1 similarly to control fibroblasts. The alternative αvβ3 integrin is almost undetectable both in control and patients’ cells. Scale bars: 10 μm.

**Table 1 genes-10-00843-t001:** Summary of clinical features of patients with biallelic *TNXB* variants.

	Present Patient	clEDS*	Total (%)
***Major criteria***			
Skin hyperextensibility (with velvety skin and absence of atrophic scarring)	+	19/19	20/20 (100)
**GJH (Beighton Score > 5/9)** (+/− recurrent dislocation)	−	15/18	15/19 (79)
**Easy bruising**	+	19/19	20/20 (100)
*Minor Criteria*			
**Articular (sub)luxation**	+	17/18	18/19 (95)
**Feet deformities** (broad/short feet, brachydactyly, pes planus, hallux valgus)	+	14/18	15/19 (79)
**Hand deformities** (acrogeric hands, mallet fingers, clinodactyly and brachydactyly)	+	4/17	5/18 (28)
**Mild proximal and distal muscle weakness**	+	9/19	10/20 (50)
**Polyneuropathy**	NA	7/18	7/18 (39)
**Edema in the legs in absence of cardiac failure**	+	4/17	5/18 (28)
Other features			
Congenital joint dislocation	+	1/17	2/18 (11)
Delayed wound healing	+	7/18	8/19 (42)
Vaginal/uterus/rectal prolapse	−	5/17	5/18 (28)
Inguinal/umbilical/wound herniation	+	4/19	5/20 (25)
Varicose veins	+	3/17	4/18 (22)
Piezogenic papulae	+	13/18	14/19 (74)
High arched/narrow palate +/− dental crowding	+	4/17	5/18 (28)
Refractive error	+	8/17	9/18 (50)
Mitral valve abnormalities	−	4/19	4/20 (20)
Diverticulosis/diverticulitis	+	4/18	5/19 (26)
Spontaneous bowel perforation	+	1/18	2/19 (11)

Note: 19 out of 30 patients reported in literature were included. Patient 3 reported in Schalkwijk et al. [9], the index patient reported in Burch et al. [8], the three patients published in Chen et al. [25] were excluded because of concomitant congenital adrenal hyperplasia. The patient described in Mackenroth et al. [26] was excluded because of concomitant *COL1A1* mutation. Patient 4 and 5 reported in Schalkwijk et al. [9] were excluded because a lack of clinical data. The patient described in O’Connell et al. [24] and the two patients described by Lindor et al. [18] were excluded because no mutation analysis was performed.

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
