# Peer review of "Clinical and Molecular Characterization of Classical-Like Ehlers-Danlos Syndrome Due to a Novel TNXB Variant"

_genes, 2019, doi:10.3390/genes10110843_

Round 1

Reviewer 1 Report

I read the manuscript “Clinical and molecular characterization of classical-like Ehlers-Danlos syndrome due to a novel TNXBvariant” by Rymen et al. The manuscript describe a patient with TNX-related EDS (cl-EDS), gives a literature review, and presents some of the changes in the ECM produced by the patient fibroblasts (decrease in COLL3, COLL5, and alpha2beta1 integrin as well as some decrease in COLL1). 

The manuscript is well written and gives a good summary of the literature.  

My only comments are the following: 

1) As a reader I would expect to get some clear introduction to the findings in the mouse model as was described by Mao et al. (this can be concluded from the text but is not sufficiently clear). 

2) "Normal relative proportions of COLLI, COLLIII and COLLV were present in the cell lysate and in the medium (data not shown)". Can authors please add these results to the supplementary data? I believe this can be a nice conformation for the finding in the mouse model showing no problem with production or secretion but in ECM organization.  

3) Figure 4, COLL1 immunofluorescence panel. Although one might get the impression of some reduced COLL1 signal in patient fibroblasts' matrix, I would expect to see a more robust and clear figure of this difference. As immunofluorescence figures might be suggestive of a desirable phenotype only based on the area chosen, either a clearer area should be chosen or a figure in a lower magnification should be presented. 

Author Response

Dear Reviewer

We thank you for the evaluation of our manuscript. We appreciated the constructive comments that were raised and revised the manuscript according to your remarks.

We have summarized point by point how we have addressed your remarks.

Comment 1

We summarized the most important findings of the paper from Mao et al. in the introduction.

Comment 2

We have added the figure in the supplementary data.

Comment 3

We have repeated the COLLI immunofluorescence on cells treated with ascorbic acid, demonstrating a more clear difference between patient and control. Figure 4  and the experimental set-up in the method section were adjusted accordingly.

We believe that the quality of the manuscript has improved by these adjustments. We hope you will find the manuscript suitable for publication.

Sincerely yours

Daisy Rymen

Reviewer 2 Report

This is a clearly presented diagnostic study on a patient with classical-like EDS caused by a novel homozygous TNXB mutation. In addition to the mutation detection, there is a detailed clinical description, and an immunostaining study using patient and control fibroblasts. A small comment - there is no mention of adding sodium ascorbate to the culture medium and based on the lack of collagen I secretion and fibril formation I'm assuming the experiments were done in the absence of ascorbate. This should be noted because in the presence of ascorbate, collagen I is efficiently secreted and fibroblasts deposit an extensive fibrillar matrix. The other collagen types are not as dependent on ascorbate for secretion. The authors bring together the published clinical and genetic information from the other published cases of TNXB mutations and identify unifying clinical features and outcomes that will be useful for clinicians to consider when managing patients. 

Author Response

Dear Reviewer

We thank you for the evaluation of our manuscript. We appreciated the constructive comments that were raised and revised the manuscript according to your remarks.

We have repeated the COLLI immunofluorescence on cells treated with ascorbic acid, demonstrating a more clear difference between patient and control. Figure 4  and the experimental set-up in the method section were adjusted accordingly.

We believe that the quality of the manuscript has improved by these adjustments. We hope you will find the manuscript suitable for publication.

Sincerely yours

Daisy Rymen